# Design and Field Validation of a Low Power Wireless Sensor Node for Structural Health Monitoring [note 1]

**DOI:** 10.3390/s21041050

**Published:** 2021-02-04

**Authors:** Federico Zanelli, Francesco Castelli-Dezza, Davide Tarsitano, Marco Mauri, Maria Laura Bacci, Giorgio Diana

**Affiliations:** Department of Mechanical Engineering, Politecnico di Milano, 20156 Milan, Italy; francesco.castellidezza@polimi.it (F.C.-D.); davide.tarsitano@polimi.it (D.T.); marco.mauri@polimi.it (M.M.); marialaura.bacci@polimi.it (M.L.B.); giorgio.diana@polimi.it (G.D.)

**Keywords:** wireless sensor node, accelerometer, structural health monitoring, energy harvesting, low power

## Abstract

Smart monitoring systems are currently gaining more attention and are being employed in several technological areas. These devices are particularly appreciated in the structural field, where the collected data are used with purposes of real time alarm generation and remaining fatigue life estimation. Furthermore, monitoring systems allow one to take advantage of predictive maintenance logics that are nowadays essential tools for mechanical and civil structures. In this context, a smart wireless node has been designed and developed. The sensor node main tasks are to carry out accelerometric measurements, to process data on-board, and to send wirelessly synthetic information. A deep analysis of the design stage is carried out, both in terms of hardware and software development. A key role is played by energy harvesting integrated in the device, which represents a peculiar feature and it is thanks to this solution and to the adoption of low power components that the node is essentially autonomous from an energy point of view. Some prototypes have been assembled and tested in a laboratory in order to check the design features. Finally, a field test on a real structure under extreme weather conditions has been performed in order to assess the accuracy and reliability of the sensors.

## 1. Introduction

Several types of structures like buildings, bridges, wind turbines, and others are subject to harsh loading scenarios and severe environmental conditions not foreseen during the design stage. Moreover, in many countries the structural and infrastructural heritage is rapidly aging and a good part of it has exceeded the design expected life. In this context, Structural Health Monitoring (SHM), which represents a fairly new concept in civil engineering, aims to monitor structural performances under service loads and to identify deterioration and damages in order to obtain the full picture of structural health. As a consequence, SHM is playing an increasingly important role in this field since the ability to continuously monitor structures can provide increased safety by means of real time warnings. In addition, the possibility to detect damage at an early stage can reduce costs and operation time, especially in avoiding periodic structural inspections, through predictive maintenance. Traditional monitoring systems make use of coaxial cables to guarantee the reliability of measured data, which ensures efficient communication on the one hand but, on the other hand, the installation and maintenance of wired devices can be very expensive in terms of time and costs. The shift of the research in the SHM field from traditional wired systems towards the use of Wireless Smart Sensors (WSS) has also been motivated by other attractive features, such as the wireless communication, on-board computation, low cost, small size, and performances nearly equivalent to wired sensors. Furthermore, the reduction in cost and size of wireless sensors allows an important measurement redundancy, with the deployment of several nodes in the structure subject to monitoring [1,2,3,4,5,6]. In this framework, the sensor node object of the present work has been developed at the Mechanical Department of Politecnico di Milano, named “WindNode”, since it has been created with the aim of performing SHM through acceleration measurements on structures mainly subject to wind-induced vibrations.

SHM aims to carry out a diagnosis of the different parts composing a structure in every moment of its life and one way to reach this task is to perform dynamic monitoring [7]. In particular, acceleration represents one of the most important parameters for SHM vibration-based methods, which have the goal of detecting structural damage through the measure of a change in the dynamic behavior of a structure [8]. Examples of SHM systems implementing dynamic monitoring concepts can be found in [9,10,11]. In order to guarantee efficient continuous vibration monitoring of structures, wireless sensor nodes have to be provided with components and features that can ensure performances adequate for the proposed task.

Firstly, concerning the measurement task, nodes have to be equipped with accelerometers that are able to acquire signals characterized by small acceleration amplitudes (a very small noise density is required in this case) and by a low frequency range (0–40 Hz), which is typical of first modes natural frequencies of many mechanical structures. In [4], the choice of a suitable MEMS (Micro Electro-Mechanical System) accelerometer for SHM purposes is widely described, with a complete comparison between the “Xnode” developed by the authors and a reference wired accelerometer. Over the same architecture of “Xnode”, a different choice of the accelerometer is carried out in [12] in order to develop a WSS suitable for sudden event monitoring (i.e., earthquake).

The wireless communication is clearly the essential element necessary to overcome the absence of cables and it has to guarantee a proper data rate while keeping the power for data transfer at a low level. A smart wireless system for the vibration monitoring of railway catenary, based on a custom-designed 2.4 GHz radio network, is shown in [13]. Time histories of several train passages are acquired during its application in a real field test, proving the efficiency of wireless communication in a case where wired systems are difficult to be used because of the necessity of insulation devices. In [14], a wireless monitoring system has been adopted successfully in the field of suspension bridges in order to provide a much more efficient and economical solution with respect to wired systems.

Lastly, the power supply aspect represents a crucial issue in the development of WSS. In fact, in most cases is not possible to cover sufficiently long monitoring time windows due to the limited autonomy provided by batteries, whose recharge implies sensors dismantling. For this reason, the growing idea in this field is to harvest energy from the environment in order to recharge batteries when an energy excess is produced. Different approaches and devices that are able to perform energy harvesting are present in the literature. In [15], the authors propose a wind energy harvester as the power source for a Wireless Sensor Network. In particular, they developed a piezoelectric impact-based micro wind energy harvester specifically designed and optimized for this task. A very interesting and peculiar solution can be found in [16], where it explains how to obtain a self-powered sensor in sea-crossing bridges by means of an oscillating buoy energy harvester (WEH). Vibrations represent one of the most attractive sources for energy harvesting, especially in situations where they are continuously generated on the monitored structure. This is the case of the solution proposed in [17] for a train wireless monitoring system, which is composed by sensors self-powered by means of maglev porous nanogenerators. More traditional solutions based on Photovoltaic (PV) panels are instead analyzed in [18,19], which demonstrate how solar energy harvesting is a reliable technology widely available in outdoor scenarios and how it allows one to achieve the highest power density compared to all other renewable sources. For these reasons, in the case of WindNode, it has been chosen to take advantage of solar energy bearing in mind that PV panels are nowadays a cheap and light technology and that they need very small preservations to last for several years.

As a result, the WindNode has been developed in order to be energetically autonomous through a balance between very low mean power consumption and the inflow energy coming from the PV panel. Low energy consumption (shown in Section 3.1.2) has been obtained both by choosing suitable electronic components and by implementing a state machine according to which the node stays on sleep mode for most of the time. This accurate consumption management strategy has been validated by means of an intensive monitoring activity of discharging and recharging phases. In addition to these advantages, the wireless communication, based on the Bluetooth Low Energy (BLE) protocol, and the accurate choice of the accelerometer allow WindNode to perform optimally continuous monitoring on real structures, as shown in detail later in the paper. The ability to carry out on board computations, such as the Fast Fourier Transform (FFT), in order to send significant and synthetic information remotely, allowing huge saving on power consumptions and avoiding to post process remotely a big quantity of data, represents clearly a great advantage in some specific applications, as in the case of the proposed field test. A brief comparison between the WindNode performances and other already developed WSS described in the literature is presented in Table 1.

## 2. WindNode Overview

### 2.1. Hardware

The developed WindNode is a WSS designed in order to perform accelerometric measurements on structures subject to wind-induced vibrations. Nonetheless, its range of application could be extended to any mechanical or civil structure affected by different sources of vibrations thanks to its inherent versatility. The WindNode is part of a wireless monitoring system composed by a certain number of nodes, depending on the specific application, and by a gateway, whose role is to receive and store all data coming from the nodes and to send them remotely using the GSM connection (Figure 1).

The communication protocol chosen for the WindNode is the BLE and this choice is driven by the excellent trade-off between long communication range and energy consumption that this protocol offers. The acquisition task is carried out by the on-board MEMS accelerometer, which allows one to provide reliable measurements together with low power consumption and very compact dimensions. Data acquired are then processed on-board by performing the FFT and synthetic data regarding harmonic oscillations with the greatest amplitudes are then transferred to the gateway using the BLE communication. The WindNode is fed by a Lithium Polimer (Li-Po) battery that is recharged by a mini PV panel. The PV panel adopted is a monocrystalline silicon panel which is characterized by a conversion rate more than 17% higher than traditional polycrystalline panels. The main features of the PV panel are reported in Table 2.

The PV panel is mounted on the top of the sensor case, which is constituted by a 3D printed body specifically designed for this application in order to host optimally all the components. The choice of the material has fallen onto Chlorinated Polyethylene (CPE), which represents a plastic material that is robust against shocks and environmental agents.

A specific circuit has been designed for on board energy management. The main function of this circuit is to supply the sensors, the microprocessor, and the wireless transmission system; the energy comes from the battery during the night and from the PV panel during daytime. Furthermore, the circuit stores the extra power produced by the PV panel (with respect to the load) into the battery. A functional scheme of the board layout is reported in Figure 2.

The power source coming from the PV panel is connected to the Integrated Circuit (IC), which is represented by the LTC3331 that performs on board power management. This component is substantially a Nanopower Buck-Boost DC/DC with an Energy Harvesting Battery Charger, which manages the battery charge using solar energy and provides a stabilized voltage supply for the microprocessor and for all on board sensors. The LTC3331 chip does not allow a Maximum Power Point Tracking (MPPT), hence it has been used with the internal hysteretic controller with a proper selection of the voltage thresholds. A Li-Po battery is chosen for this device and this technology, in fact, allows one to have the highest level of energy density and hence it is possible to minimize the battery size both in terms of dimensions and weight. The main features of the battery are reported in Table 3.

In order to continuously monitor the battery status of the node, it has been decided to adopt a battery gauge, as can be seen in Figure 2. A proper IC has been chosen for this task and is represented by the LTC2942. This circuit allows one to have an estimation of the battery State of Charge (SoC) through a Coulomb Counter, to measure the battery voltage and temperature in the IC and to have a very high accuracy both on voltage and charge measurements. The other sensors present on the board are a MEMS triaxial accelerometer, with embedded sensor conditioning and Analog to Digital Converter (ADC), and a temperature sensor. An accelerometer suitable for SHM purposes has been found in the Analog Devices ADXL 345, thanks to some attractive features such as ultra-low power consumption, user selectable amplitude g-range, and wide operating temperature range (Table 4).

Regarding the temperature sensor, an additional thermal probe with respect to the one embedded in the battery gauge IC has been added in order to be able to measure the temperature in a specific point of the board or far from the Printed Circuit Board (PCB) through proper wiring. The temperature sensor is constituted by a Negative Temperature Coefficient (NTC) resistor, and by using an appropriate voltage divider the resistance value of the NTC resistor can be computed by the microprocessor, which is represented by the DSPIC33EP512GP806-I/PT from Microchip.

All the measurements carried out by means of the different sensors are managed on the microprocessor using high speed serial data protocols, as Inter-Integrated Circuit (I2C) or Serial Peripheral Interface (SPI), or using directly the microprocessor ADC. Lastly, after having analyzed the collected data, the microprocessor is able to transmit some synthetic values to the gateway using the wireless transmission system. The device chosen to perform this task is the Fanstel 2.4 GHz BT840F RF Transceiver (based on Nordic nRF52840) which is able to support BLE and transmit data, using this communication protocol, in the range of a few hundred meters as it will be shown in Section 3.1.4. A 3D rendering of the designed board and an external view of the first prototype of WindNode are visible respectively in Figure 3a,b [1].

### 2.2. Software

The on board firmware has been specifically developed for this project and is divided between the two available processors, namely the sensor node microprocessor, whose role is to perform the computational work, and the CPU inside the BLE transceiver, whose task is to manage data communication. The node operativity is characterized by an optimized duty cycle composed by a sleep period and a working one. During the sleep time, which can be increased or decreased according to the monitoring requirements by modifying the node wake time value via software, the Microchip microprocessor is in a very low power mode while the communication module is put in a reset state. Under these conditions the power consumption is very low, as it will be shown in Section 3.1.2. The communication between sensor nodes and gateway is realized by means of messages exchange between the two devices. In particular, the state machine shown in Figure 4 represents the operations performed by WindNodes in one acquisition cycle in the specific case of wind-induced vibration monitoring on cables.

At first, the gateway waits until it recognizes that each sensor node has waked up and sends an acknowledge in response to every wake message received in order to put WindNodes in idle state. After this operation, the gateway sends a start message to one node at a time in order to start an acquisition in one of the two implemented working mode (Aeolian or Subspan) and waits for the node answer. The WindNode acquires the signals from Y and Z axes of the accelerometer and computes data by performing the FFT over a limited frequency range, i.e., 0–100 Hz. The resulting maximum harmonic amplitude and the related frequency are then transmitted by each node to the gateway. The gateway looks for the maximum amplitude and its related frequency among all data received from WindNodes and it asks to each sensor the spectrum line corresponding to that frequency (as well as the adjacent lines). This operation is performed in order to get the information about the most excited mode, which is the one that mainly contribute to fatigue issues on the conductor. In addition, more nodes suitably positioned on the structure to be monitored give the possibility of reconstructing that modal shape. In the end, after the last acknowledge, sensor nodes are put in a sleep state for a chosen amount of time and once this time has elapsed, another acquisition cycle begins. Furthermore, the WindNode is able to exchange information with the gateway also to modify acquisition parameters such as sampling frequency, full scale range, wake time, etc.

## 3. Experimental Validation

The performance of the developed sensor node has been evaluated experimentally by means of both laboratory tests and a field test on a real structure. On one hand, laboratory tests are fundamental in order to check the design features and to solve preliminary issues but on the other hand, only field tests in real scenarios can assure that the prototype is suitable for the task of SHM of existing structures.

### 3.1. Laboratory Tests

After the realization of the first WindNode prototype, many laboratory tests have been performed in order to assess the good overall functioning of the sensor node and its peculiar features. In this section, the four most significant ones are presented, namely the shaker test, the consumption test, the temperature test and the communication range test.

#### 3.1.1. Shaker Test

A shaker test was carried out in order to evaluate the measurement accuracy of WindNode through a comparison with a reference wired accelerometer (Figure 5a). A second reference accelerometer was used in order to verify the measurement chain, by detecting potential local modes involving the board and plastic enclosure. The test set-up is composed by a LDS electromagnetic shaker, a 33220A Agilent waveform generator, and a PA100E LDS power amplifier, as observed in Figure 5b. Data were acquired wirelessly from the WindNode, while in the case of the reference accelerometer a DAQ 9178 National Instruments was used. This test was performed by means of a frequency sweep and a few acquisitions were carried out for each considered frequency.

The transfer function between the sensor node and reference accelerometer was evaluated by performing the FFT and by taking the spectrum lines corresponding to the forcing frequency. In Figure 6a–c the transfer functions of x, y, and z axes are represented [1]. The expected value is unitary over the whole frequency range tested and is represented by the black line. Since the excitation input was a monoharmonic sine, WindNode acquisitions were affected by leakage issues because the power distributes itself on the spectrum adjacent lines (represented with the blue line). This can be explained firstly because the WindNode samples on a non-integer number of periods (the acquisition number of points is in fact fixed to 512 due to the available hardware and software resources) and secondly because a variation up to 20% on the chosen sampling frequency was observed. Moreover the sampling frequency is driven by the accelerometer chip and cannot be modified. In order to overcome this time window length issue, a correction on the measured amplitude has been applied according to (1), by taking the RMS (Root Mean Square) of the signal frequency band around the peak and obtaining the corrected amplitude A.
(1)$$A=\sqrt{{S(i}_{\max}{-1)}^{2}{+S(i}_{\max}{)}^{2}{+S(i}_{\max}{+1)}^{2}}$$

In (1), S represents the Discrete Fourier Transform amplitude while i_max_ is the spectrum line corresponding to the maximum amplitude. This equation allows one to take into account, following an energetical interpretation, the energy spread due to leakage on spectrum lines adjacent to the one corresponding to the maximum amplitude detected. On the other hand, the reference accelerometer has not been affected by leakage since the sampling is performed at a very high frequency (25 kHz) and since the acquisition window time length has been chosen in order to obtain an integer multiple of the oscillation period. Looking at the obtained red line, it can be observed that the transfer function trend is quite coherent and sufficiently flat over the entire considered frequency range. It can be noted that the WindNode offers a global underestimation of the acceleration by a 10% mean factor, therefore this calibration procedure is essential in order to compensate data acquired on the field.

In order to validate the test procedure, an analysis of the measurement uncertainty in the accelerometer calibration was carried out according to the GUM Framework [20]. Since a few acquisitions with different input acceleration amplitudes were performed for each tested frequency, the standard uncertainty u_c_ was estimated as the standard deviation for each of these amplitudes. Then, the expanded uncertainty was determined according to (2):U = k u_c_(2)

In (2), a value of the coverage factor k = 2 is used. At this point, error bands for each frequency was computed and in this last step only the uncertainty corresponding to the maximum amplitude tested for each frequency was considered as a reference. In fact, due to shaker technical limitations, it has been possible to use only very small amplitudes as input for low frequencies, taking into account the lowest values among them would have led to meaningless uncertainty values. In Figure 6d, it is clearly visible how the uncertainty is higher, as it was expected, in the low frequencies range (below 10 Hz), since in those cases the small input amplitudes employed were close to the WindNode amplitude resolution. Uncertainty values are smaller for higher frequencies, allowing one to consider the presented testing procedure as a good benchmark to evaluate the measurement performance of the developed sensor node.

#### 3.1.2. Consumption Test

The second laboratory test regards the verification of the sensor node power consumption during the working phase, since this feature represents a key point in the design of the device. Apart from minimizing the power consumption of the electronic components during the acquisition and transmission phases, the design optimization was performed by implementing a sleep mode between two consecutive measurement operations. Consequently, it is possible to obtain a lower mean power consumption by increasing the WindNode sleep time and this operation can be executed through the acquisition software. The obtained duty cycle is in line with continuous monitoring of civil and mechanical structures and allows to gather a proper amount of data with a reasonable autonomy in case of no solar light hitting the PV panel. The test set-up is represented in Figure 7a. The performed simulations allows to collect consumption data, represented in Figure 7b in terms of current consumption (blue line) of a working WindNode characterized by a 30 s wake time duty cycle [1].

In the node sleep time condition, the mean current consumption *I_mean Sleep_*, identified with the red line, is very low. The black line represents the mean current consumption *I_mean Acq-Transm_* during the data acquisition and transmission phase, which is clearly the most expensive one in these terms. Considering the implemented duty cycle, it is possible to compute the mean current *I_mean Cycle_* used by the sensor node during one measurement cycle (which is the cycle time, corresponding to the sum of the sleep and acquisition time) through (3):(3)$$I_{mean\;Cycle}=\;\frac{\int_{0}^{t_{Sleep}}I_{Sleep}\;dt+\int_{0}^{t_{Acq}}I_{Acq}\;dt}{t_{Cycle}}$$

In this way, the effective current consumption of the sensor node, represented by the green line in Figure 7b, has been obtained. Numerical results of this test are summarized in Table 5. Assuming that the PV panel is for whatever reason not able to recharge the battery at all, the WindNode could still be able to perform continuous measurements for more than one month thanks to its very low consumption.

#### 3.1.3. Temperature Test

In order to test the device functionality in extreme conditions, which were expected in its first use on the field in Manitoba (Canada), a temperature test was arranged. In particular, the performance of WindNode was evaluated by positioning the sensor node into a freezer capable of reaching approximately −20 °C. A professional temperature probe (usually employed for air temperature calibration in the Wind Tunnel of Politecnico di Milano) was used in order to have a reliable measurement inside the freezer during the test. It must be noted that the battery equipping WindNode in this test is of the same type described in Section 2.1 but with a capacity of 1300 mAh. Two kind of tests were carried out:
Test 1. Sensor node with battery discharging;Test 2. Sensor node with battery linked to a power supply in order to emulate the PV generation.


In Test 1, the WindNode was turned on and placed in the freezer. Firstly, it was checked that the sensor node was able to communicate correctly, then the battery performance in relation to temperature was monitored. As observed in Figure 8a, the battery equipping the WindNode was not completely charged at the beginning of the test. In the presence of a mean temperature of −19 °C, a fast discharge of the battery (approximately 0.75 mAh per second, equivalent to 2.78 mW) was observed, until the SoC reached the zero value after 15 min. The test duration was extended to 30 min in order to monitor the battery voltage trend.

In Test 2, the solar energy coming from the PV panel was simulated by means of a power supply connected to the node with a 5.5 V level. The sensor node communication was checked and the ability of the external power source to recharge the battery also in presence of a very low temperature (approximately −16 °C) was verified. In particular, after 10 min the SoC value was approximately 10% of the total capacity, as it can be appreciated in Figure 8b. Once it was verified that the battery recharge was performing effectively, the power supply was removed and a battery discharge rate, very similar to the one detected in Test 1, was observed.

#### 3.1.4. Communication Range Test

In the end, a transmission check was carried out in order to quantify the communication range of the developed sensor node in a real-life scenario. The Bluetooth transceiver mounted on the WindNode was declared by the manufacturer to be able to communicate up to a distance of 2 km on sight at a height of about 1.5 m over the ground. However, since this kind of testing is usually performed in an open and clear environment (e.g., the desert), this information cannot be considered representative of the sensor node performances in a typical monitoring context. For this reason, the experimental test was performed in an urban scenario, visible in Figure 9a, for increasing distances between the sensor node and master board, represented by a Fanstel BT840 evaluation board (EV-BT840E) equipped with an ANT060 external antenna [1]. Three stations were chosen along the path as reference points for communication measurements with the distances specified in Table 6. During the test, the master board and WindNode positioning are indicated in Figure 9b. Several acquisition cycles were performed in each station in order to verify the robustness of the data reception.

In addition to the consistency of the data received, the parameter taken into account as the indicator of the communication quality between the two devices is the Received Signal Strength Indicator (RSSI). The RSSI is a metric that indicates the power of the received radio signal [21]. Measurement techniques taking advantage of this parameter rely on the principle that the radio signal is increasingly attenuated as the distance between the two devices increases [22]. In addition, the external environment complexity causes the signal weakening during its propagation and, overall, the greater the propagation distance is, the greater the signal attenuation is. The RSSI is expressed in dBm and its value can range from −100 dBm to 0. The closer the value is to 0, the stronger the received signal is however, values between −30 dBm and −80 dBm are still signs of excellent communication. Getting closer to −90 dBm, the communication quality starts to decrease and therefore this value can be taken as threshold for very difficult communication or no communication at all. As seen in Table 6, the outcome of the test is that the WindNode communication range is around 200 m. This result is coherent with such a harsh scenario, where many reflections and interferences are present.

If it is necessary for some specific applications, the communication range could be extended by adopting an external antenna instead of the planar one realized on the Bluetooth PCB transceiver. From this perspective, a test similar to the one presented has been performed in a different scenario (countryside scenario, Figure 10a) in order to evaluate the performance of a WindNode equipped with a compact omni-directional antenna, visible in Figure 10b.

As observed in Table 7, the communication range has increased with respect to the previous case and it could even be enhanced through deeper studies on antenna positioning.

However, in this configuration the antenna is clearly not enclosed in the sensor case and therefore it can be damaged by debris or by meteorological actions. For this reason, since the communication range test in the urban scenario has allowed to carry out a satisfying result with respect to the requirements, it has been preferred to keep the original design of the sensor node.

### 3.2. On-Site Experimental Campaign

In order to test WindNode performances, a quite long experimental campaign was arranged. Beside the measurement efficiency of the sensor, the purpose of this experimental test was to prove that the WindNode was also able to work properly in real operative conditions, which represents a completely different scenario with respect to a laboratory environment. In this context, the main threats for the “good health” of the sensor are represented by difficult weather conditions. Environmental temperature, in particular, represents a critical parameter that could give issues to the electronic components present on the board and that could make it even impossible to recharge Lithium batteries. Another goal was to test the efficiency of the PV panel used to recharge the sensor node battery in real conditions, in order to validate this power supply solution in the context of structural monitoring in an open environment.

#### Manitoba Field Test

The experimental campaign took place in Manitoba (Canada) in February 2019, in the context of the monitoring of wind-induced vibrations on a High Voltage Transmission Line, which represents a well-known but still challenging field for monitoring applications [23,24]. In this case, the goal was to perform an effectiveness check of the dampers mounted on the conductors to mitigate the phenomenon of Aeolian Vibrations. This kind of instability is mainly due to the shedding of wind-induced vortices from the conductor, which creates an alternating pressure unbalance giving rise to forces able to move the conductor up and down [25]. Aeolian Vibrations are generated by a moderate wind (0.8 to 7 m/s) and typical aeolian vibration frequencies lie between 4 and 120 Hz [26]. The frequency is given by the Strouhal formula (4):(4)$$f\;=\;S\;\frac{V}{D}$$

In (4), V is the wind velocity, D is the conductor diameter and S is the Strouhal number, which can be taken equal to S = 0.18 for a circular shape. In Figure 11a, three WindNodes mounted on the lower conductor are visible [1]. The campaign duration was about 3 months and useful data were collected. Moreover, the cold Canadian winter has represented a tough benchmark for the WindNode electronic functionality. In particular, the few hours of daylight and very low temperature put stress on the battery and energy harvester efficiency. The battery behavior of one of the sensor node installed on the conductor over a 5-day time-window is shown in Figure 11b. It can be observed that the battery voltage trend follows the environmental temperature one. The voltage, in fact, shows a drop during night hours when no sun is present and temperature decreases up to −20 °C, which represents a critical value for Li-Po battery health. Nevertheless, when the PV panel is hit by the sunlight, the battery is also recharged in the presence of low temperatures, as it can be appreciated in the graph.

As already explained, data output coming from WindNodes are in the form of vibration amplitudes and frequencies, thanks to the ability of performing the FFT of the acquired acceleration signal on board. In this way, synthetic and significant data are already available to be analyzed once received by the WindNodes. In the case of conductors, acceleration values correlate with wind data acquired by an anemometer in order to understand what kind of wind instability is affecting the line. Some data acquired from WindNodes during the Manitoba field test are shown in Figure 12a,b. In particular, in Figure 12a it is clearly visible that the detected frequency values in the range 5–50 Hz are in accordance with Strouhal formula. The accuracy of the sensor node allows one to detect acceleration amplitudes starting from the MEMS accelerometer amplitude resolution up to 0.4 g, as it can be appreciated in Figure 12b [1]. In accordance with the explained physics of the problem, the higher amplitude values correspond to low frequencies, while for relatively high frequencies (30–50 Hz range) the lowest amplitude values are observed. Data acquired during this experimental campaign on a real structure have proven the good performance of WindNode in the field of continuous monitoring of a vibrating structure.

## 4. Conclusions

The paper illustrates the development of a new WSS to be used for SHM purposes. The design of the device was deeply analyzed, highlighting the key features in terms of both hardware and software. Several laboratory validation tests were carried out in order to assess some important features, namely the good measurement accuracy, the low power consumption, the ability to withstand low temperatures, and the wide communication range. In the end, the coherence of the data acquired during a field test on a real structure allowed us to verify the good performances of WindNode and to validate its use in SHM field. In addition, the experimental campaign has proven that the sensor node can be adopted in scenarios characterized by extreme weather conditions.

## Figures and Tables

**Figure 1 sensors-21-01050-f001:**
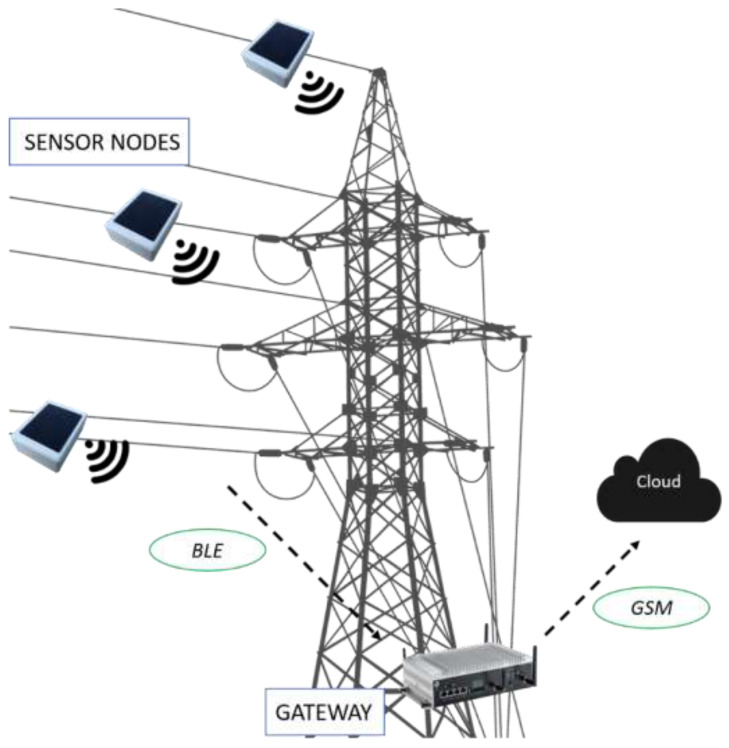
Architecture of the developed Wireless Monitoring System.

**Figure 2 sensors-21-01050-f002:**
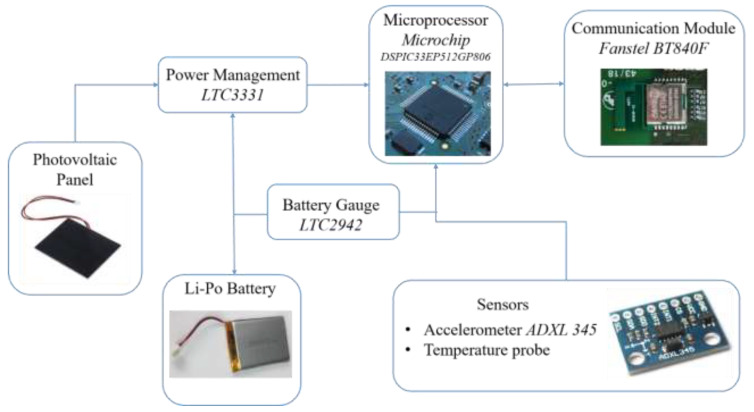
Functional scheme representing the main components present on the board.

**Figure 3 sensors-21-01050-f003:**
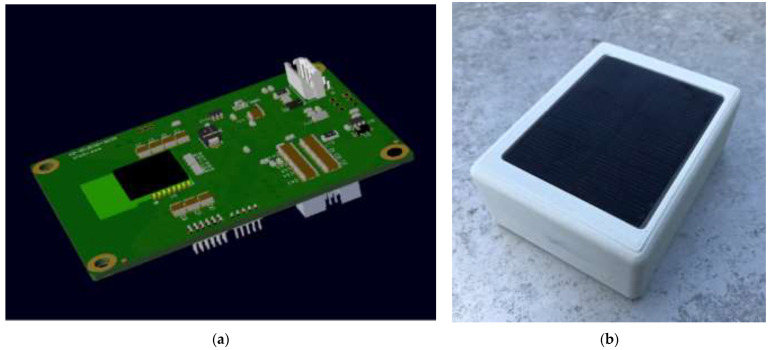
Developed sensor node: (**a**) 3D render of the designed electronic board and (**b**) first prototype of WindNode [1] © 2020 IEEE.

**Figure 4 sensors-21-01050-f004:**
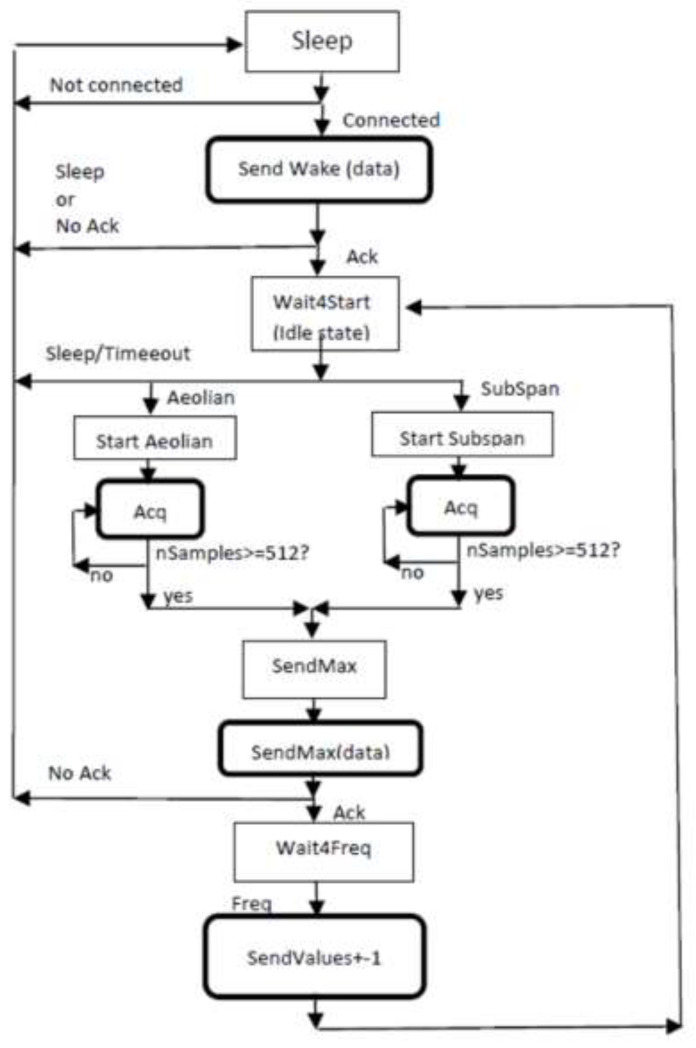
State machine characterizing the WindNode software.

**Figure 5 sensors-21-01050-f005:**
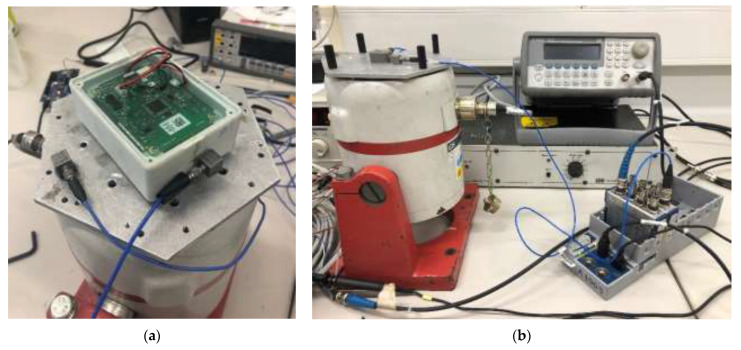
Shaker test: (**a**) WindNode and reference accelerometers mounted on the shaker and (**b**) overview of the instrumentation composing the test set-up.

**Figure 6 sensors-21-01050-f006:**
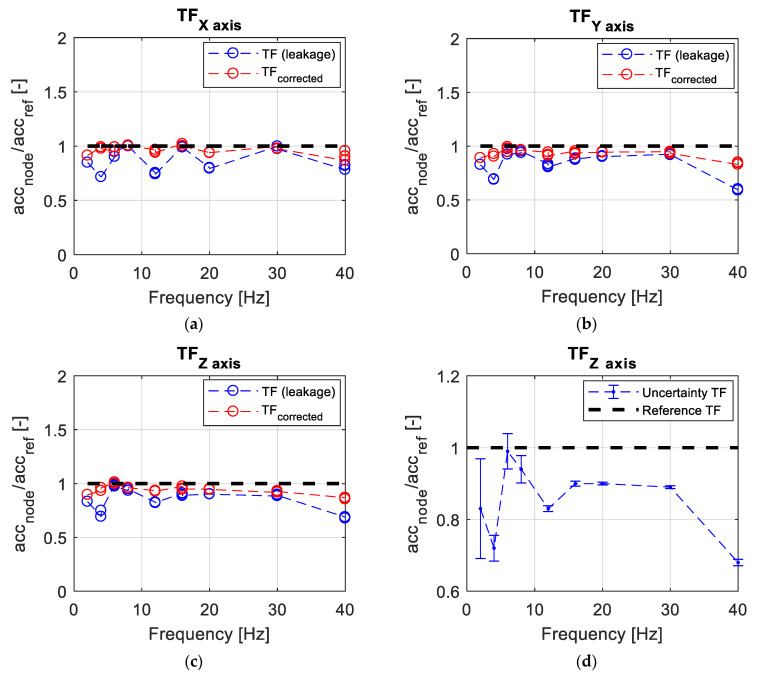
Shaker test results: (**a**) X axis transfer function; (**b**) Y axis transfer function; (**c**) Z axis transfer function; and (**d**) uncertainty estimated by means of error bands for each considered frequency [1] © 2020 IEEE.

**Figure 7 sensors-21-01050-f007:**
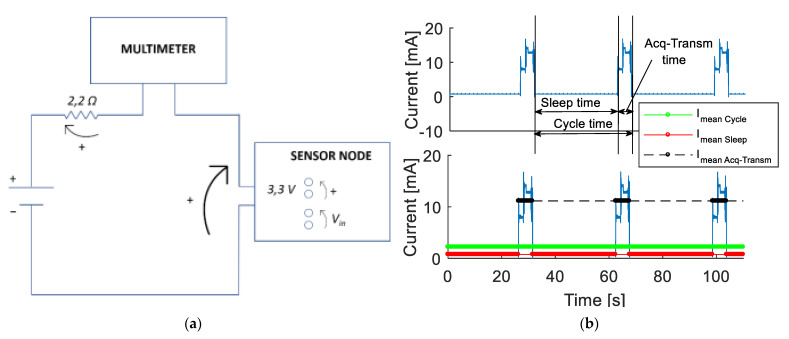
Consumption test: (**a**) Scheme representing the test set-up and (**b**) current consumption results [1] © 2020 IEEE.

**Figure 8 sensors-21-01050-f008:**
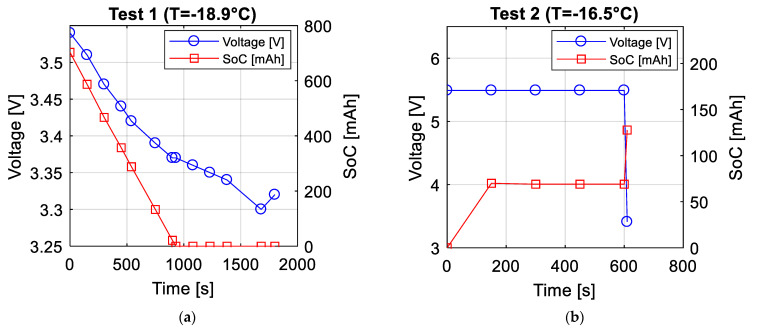
Temperature test: (**a**) Battery voltage and SoC (State of Charge) trends when battery is discharging (Test 1) and (**b**) battery voltage and SoC trends when battery is recharging (Test 2).

**Figure 9 sensors-21-01050-f009:**
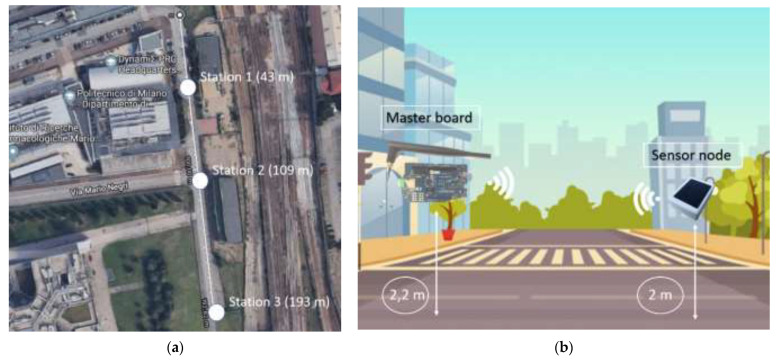
Communication range test in urban scenario: (**a**) Map of the test area [1] © 2020 IEEE and (**b**) positioning of master board and sensor node during the test.

**Figure 10 sensors-21-01050-f010:**
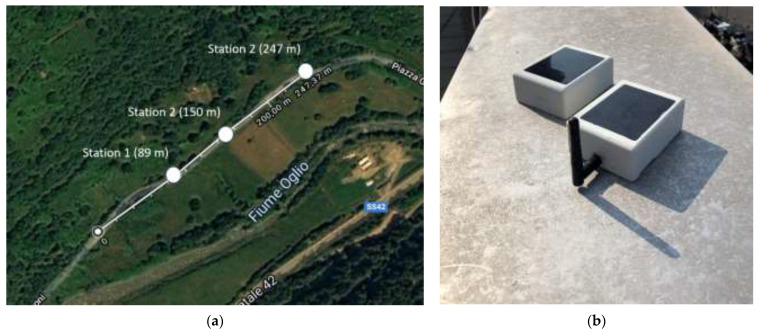
Communication range test in countryside scenario: (**a**) Map of the test area and (**b**) WindNode endowed of the external antenna.

**Figure 11 sensors-21-01050-f011:**
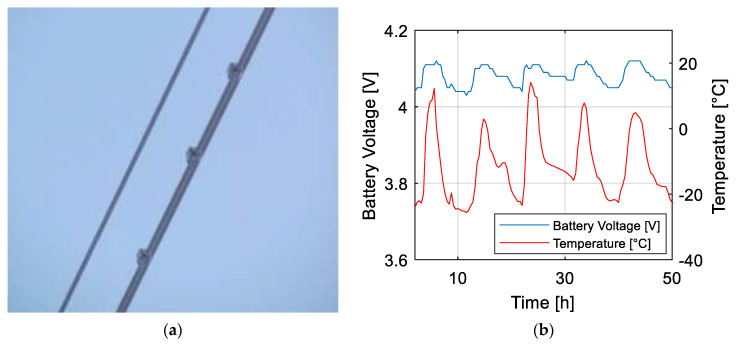
Manitoba field test: (**a**) WindNodes mounted on the lower conductor [1] © 2020 IEEE and (**b**) battery voltage trends with respect to environmental temperature.

**Figure 12 sensors-21-01050-f012:**
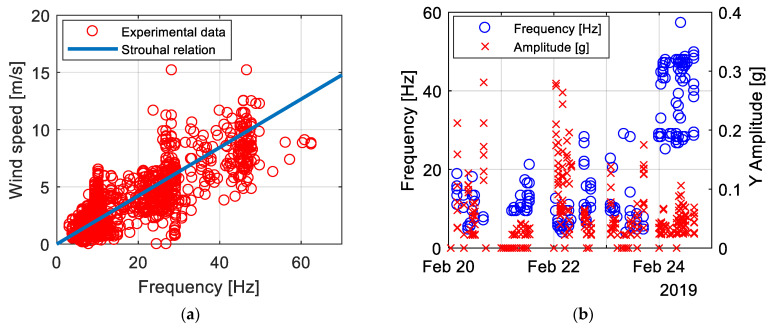
Manitoba test results: (**a**) Comparison between experimental data and Strouhal relation and (**b**) experimental data representing amplitudes and frequencies of vibration.

**Table 1 sensors-21-01050-t001:** Performances comparison between WindNode and other Wireless Smart Sensors (WSSs).

	Comm. Protocol	Transducer	Energy Harvester	Comm. Range	Other Features
WindNode	BLE	MEMS tri-axial acc.	PV Panel	<200 m	On-board FFT
Xnode in [4]	Zigbee	MEMS tri-axial acc.	External	Not declared	-
Sensor in [13]	2.4 GHz radio custom designed	MEMS gyro	-	Not declared	-
Sensor in [9]	900 MHz Radio Frequency	Monoaxial acc.	-	<300 m	On-board FFT

**Table 2 sensors-21-01050-t002:** PV panel main parameters.

Parameter	Value
Technology	Monocrystalline Silicon
Nominal Power	0.5 W
Typical Voltage	5.5 V
Typical Current	100 mA
Dimensions	70 × 55 × 3 mm

**Table 3 sensors-21-01050-t003:** Battery main parameters.

Parameter	Value
Technology	Lithium Polymer
Voltage	3.7 V
Capacity	2000 mAh
Dimensions	44 × 72 × 7 mm
Weight	40 g

**Table 4 sensors-21-01050-t004:** Accelerometer main parameters.

Parameter	Value
Measurement range	±2 g, ±4 g, ±8 g, ±16 g
Sensitivity	256 LSB/g
Noise	1.1 LSB rms
Output Data Rate	3200 Hz
Operating Voltage	3.3 V
Supply Current	140 µA
Operating Temperature Range	−40 ÷ 85 °C

**Table 5 sensors-21-01050-t005:** Consumption data results.

Mode	Mean Current Consumption [mA]	Mean Power Consumption [mW]
Sleep mode	0.7	2.6
Acquisition-Transmission	11.1	41.1
Complete cycle	2.2 ^1^	8.1 ^1^

^1^ Values computed by means of (3).

**Table 6 sensors-21-01050-t006:** Communication test results—Urban scenario.

Station Number	Station Distance [m]	Node RSSI [dBm]
1	43	−74
2	109	−75
3	193	−86

**Table 7 sensors-21-01050-t007:** Communication test results-Countryside scenario.

Station Number	Station Distance [m]	Node RSSI [dBm]
1	89	−76
2	150	−78
3	247	−88

## Data Availability

Not applicable.

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
