# Peer review of "Design and Field Validation of a Low Power Wireless Sensor Node for Structural Health Monitoring"

_sensors, 2021, doi:10.3390/s21041050_

Round 1

Reviewer 1 Report

  1. An interesting paper regarding a new WSS to be used for SHM purposes.
  2. This manuscript is an extension version of a conference paper, please describes the difference between these two papers.
  3. About temperature testing, why it tests only in two different degrees?
  4. Is there any other published WSS for SHM, I suggest author can give a brief performance comparison.  

Reviewer 2 Report

The proposed paper focuses on Design and field validation of a low power wireless sensor node for Structural Health Monitoring.

The author must perform a major revision.

  • authors must improve English
  • the authors should better explain the hardware architecture. The authors speak of microprocessor, which type is used? In line 165 the authors say to use microprocessor’s adc and in the software section they talk about firmware. Explain better and tell if you are using a microprocessor or microcontroller.
  • For the power section explain better the PV section. What features does the PV panel used in the article have? How was the LTC3331 chip set up with the used panel? What is the maximum power point?
  • In the introduction, explain the types of monitoring (static and dynamic) and add some citations. What kind of monitoring do the authors use in this paper? Examples of articles that talk about static and dynamic structural monitoring:
  1. DOI: 10.1002/stc.303
  2. DOI: 10.3390/s20174908
  3. DOI: 10.1080/15732470600590820

Reviewer 3 Report

This paper presents a wireless accelerometer mainly characterized by very low mean power consumption and the inflow energy coming from the PV panel. The performance of the sensor is verified by several experiments. It can be applied in some special scenes. For better presenting the work in the article, some suggestions are noted below.

  • Being an accelerometer, the developed WindNode adopts a power saving strategy that, however, only outputting the resulting maximum harmonic amplitudes and the related frequencies is severely limiting on the information contained in the monitoring data, which is recommended that presenting the time history data to broaden its application scenarios.
  • The reviewer is interested in how “ the resulting maximum harmonic amplitudes ” are considered. As we known, the FFT data contain different modal response with varying resonance peak. It is not trivial to determine the maximum amplitude of each mode in a spectrum.
  • To overcome the time window length issue, the authors present a correction on the measured amplitude. The reviewer wonders why the issue can’t be fixed through resetting the parameter of the WindNode, such as the acquisition number of points, sampling frequency, and why the correction could be done in this way ( eq(1) ) ?
  • The authors are suggested to illustrate the “ duty cycle “ more clearly. In figure 7b, the working WindNode characterized by a 30 seconds wake time. However, the period of blue line is obviously not 30 seconds, while the period of red line seems to be less than 30 seconds

Round 2

Reviewer 2 Report

All issues have been addressed
